# Genetics and Pathogenetic Role of Inflammasomes in Philadelphia Negative Chronic Myeloproliferative Neoplasms: A Narrative Review

**DOI:** 10.3390/ijms22020561

**Published:** 2021-01-08

**Authors:** Valeria Di Battista, Maria Teresa Bochicchio, Giulio Giordano, Mariasanta Napolitano, Alessandro Lucchesi

**Affiliations:** 1Hematology Unit, IRCCS Istituto Romagnolo per lo Studio dei Tumori (IRST) “Dino Amadori”, 47014 Meldola, Italy; valeria.dibattista@irst.emr.it; 2Biosciences Laboratory, IRCCS Istituto Romagnolo per lo Studio dei Tumori (IRST) “Dino Amadori”, 47014 Meldola, Italy; teresa.bochicchio@irst.emr.it; 3Division of Internal Medicine, Hematology Service, Regional Hospital “A. Cardarelli”, 86100 Campobasso, Italy; giuliogiordano@hotmail.com; 4Department of Health Promotion, Mother and Child Care, Internal Medicine and Medical Specialties (PROMISE), Haematology Unit, University Hospital “P. Giaccone”, University of Palermo, 90127 Palermo, Italy; mariasanta.napolitano@unipa.it

**Keywords:** inflammasome, NLRP3, AIM2, myeloproliferative neoplasms, chronic inflammation

## Abstract

The last decade has been very important for the quantity of preclinical information obtained regarding chronic myeloproliferative neoplasms (MPNs) and the following will be dedicated to the translational implications of the new biological acquisitions. The overcoming of the mechanistic model of clonal evolution and the entry of chronic inflammation and dysimmunity into the new model are the elements on which to base a part of future therapeutic strategies. The innate immune system plays a major role in this context. Protagonists of the initiation and regulation of many pathological aspects, from cytokine storms to fibrosis, the NLRP3 and AIM2 inflammasomes guide and condition the natural history of the disease. For this reason, MPNs share many biological and clinical aspects with non-neoplastic diseases, such as autoimmune disorders. Finally, cardiovascular risk and disturbances in iron metabolism and myelopoiesis are also closely linked to the role of inflammasomes. Although targeted therapies are already being tested, an increase in knowledge on the subject is desirable and potentially translates into better care for patients with MPNs.

## 1. Genetic Background: The “Mechanistic” Model of MPNs

Chronic myeloproliferative neoplasms (MPNs) are disorders of the hematopoietic stem cell, after it has undergone somatic mutations and consequent clonal expansion. The result is an uncontrolled proliferation of the figurative elements of the blood [1]. The natural history of these neoplasms is marked by frequent venous and arterial thrombosis, hemorrhages and transformation to acute myeloid leukemia (AML) [2,3]. The relationship between the genotype of these pathologies and their clinical phenotype or the risk of evolution has only been partially elucidated.

MPNs harbor somatic mutations in the *JAK2* gene (exon 12 or 14), *CALR* gene (exon 9) or *MPL* gene (exon 10). V617F mutation falls in the inhibitory pseudokinase domain of the *JAK2* gene, mutations of *MPL* fall at position W515 in the juxtamembrane domain, while pathologic *CALR* mutations are out-of-frame insertion and/or deletions generating a novel C-terminal peptide: all of them cause constitutive activation of MPL/JAK/STAT signaling axis [4,5,6]. Mutations in these genes are found to be almost entirely mutually exclusive, although rare cases have so far been reported of patients with a double mutation in *JAK2* and *CALR* [7,8]. Besides the aforementioned driver mutations, 10–15% of MPNs that do not harbor any of these common mutations are defined as triple-negative and usually show a more unfavorable prognosis [9] (Figure 1). Additional mutations, such as those in epigenetic modifiers (specifically *ASXL1, TET2* and *EZH2*), spliceosomal components (*SRSF2, U2AF1* and *SF3B1*), metabolic modifiers and linker proteins (*IDH2, SH2B3* and *CBL*), play a role in disease progression inducing increased self-renewal and block in differentiation [10]. 

These so-called “high molecular risk” (HMR) somatic mutations represent molecular markers for the identification of primary myelofibrosis (PMF) patients at high risk of death, leukemic transformation or fibrosis development [11]. A great majority of MPNs harbor these somatic mutations other than *JAK2/MPL/CALR* “driver” mutations [12]. Surprisingly, they often are already present at diagnosis and only rarely acquired during progression [13]. 

It is important to remember how—in the ten years since the discovery of *JAK2′*s “gain of function” mutations—the mechanistic model of progressive clonal expansion dominated the scientific landscape of MPN. The detection of other driver mutations and of “high-risk” genetic lesions has certainly been a giant step in the understanding of the enormous complexity underlying the pathogenesis of these diseases. The proposed model, fundamentally based on genetic instability and on the progressive acquisition of point mutations, efficiently represents the “stages” of the “biological continuum” theorized by Dameshek in the 1950s. Neither the terrain on which these stages are based, nor the vehicle to reach them, were clearly portrayed in this representation.

## 2. MPNs as Inflammatory Diseases

In 2015, precisely ten years after the identification of the *JAK2V617F* mutation, in an absolutely brilliant way, Hasselbalch and Bjørn collected the epidemiological, biological, pathogenetic and clinical evidence to consider MPN as inflammatory diseases, a paradigm of the connection between chronic inflammation and oncogenesis [14]. In the authoritative review, they also introduced the concept of immune dysregulation. The loss of anti-tumor immunological surveillance is also of fundamental importance in this context.

However, focusing on chronic inflammation, it is universally considered one of the chief promoters of vascular damage and, in particular, of endothelial damage. Driver mutations, especially in *JAK2* gene, seem to be the initiators of the inflammatory state of the vascular system. It has been shown that an increased expression of the *JAK2V617F* mutation on endothelial cells (ECs) of patients with MPN is responsible for functional alterations: the vascular bed is more inflamed and permeable, there is limited cell growth and a more rapid senescence [15]. 

The link between inflammation, vascular damage and dysimmunity is very strong, so much so as to represent a second “biological continuum” in addition to the one underlying the genotypic and phenotypic evolution of the disease. The role of the innate immune system and the balance between anti- and pro-inflammatory cytokines on endothelial function has been known for some time, thanks to models that are to be considered transversal in the context of various chronic (even non-neoplastic) diseases [16,17]. Being affected by a chronic myeloproliferative neoplasm is therefore an undeniable reason for the acceleration of atherosclerosis processes, especially if in the presence of additional cardiovascular risk factors. In addition, if on the one hand it strongly increases the probability of incurring thrombotic or hemorrhagic events, on the other the cytokine storm, overwhelming manifestation of inflammation and dysimmunity modifies the oncological history.

Immune dysregulation in cancer has become an active field of investigation for its ability to create a permissive microenvironment enabling immunosurveillance escape and tumor growth. Non-neoplastic chronic inflammation, secondary to chronic infections or autoimmune disorders, provides a continuous release of pro-inflammatory cytokines such as TNF-α, IL-6, IL-8 and accumulation of reactive oxygen species (ROS), which in turn sustains cancer development and progression through genetic instability, oxidative stress, inhibition of apoptosis program and cell migration [18].

In this context, MPNs represent ideal “inflammatory” disease and a useful model to assess the relationship between chronic inflammation, loss of immune tolerance and clonal proliferation. In PMF, specifically, chronic inflammation plays a central role in the pathogenesis of the disease since the cross-talk between malignant hematopoietic cells and normal stromal cells determines the clonal progression through release of pro-inflammatory cytokines and chemokines [14,19,20]. MPN-specific oncogene mutations such as *JAK2V617F* and *MPLW515L* stimulate the JAK/STAT3 pathway to enhance inflammatory cytokine production by autocrine and paracrine mechanisms, promoting the growth of the other MPN cells but suppressing the growth of normal cells. JAKs are critical for the signaling of many surface cytokines and growth factor receptors [21]. After binding of specific cytokines, JAKs undergo transphosphorylation, then phosphorylate tyrosine residues on the receptor, creating a SH2 docking site for the STAT transcription factors, which transmit the cytokine activation signals from the cytoplasm into the nucleus and induce the transcription of downstream targets involved in cell growth, differentiation, and apoptosis [22,23]. STATs target genes include cytokines and other growth factors that in turn activate JAK/STAT signaling creating a potential autocrine positive feedback loop [24]. The importance of this pathway has been also provided from the evidence that even the other somatic mutations (mutations in exon 12 of *JAK2* and gain-of-function mutations in *MPL*) detected in MPNs are JAK-activating mutations [25]. 

Notably, regardless of the *JAK2V617F* mutation, an overrepresentation of the JAK2 46/1 haplotype (characterized by rs3780367, rs10974944, rs12343867 and rs1159782 polymorphisms) is described in MPNs. It confers increased risk of more severe inflammatory response and is associated with myelofibrotic transformation and shorter overall survival [26,27].

In MPNs, many cytokines are upregulated, creating a sustained inflammatory microenvironment that correlates with more severe marrow fibrosis and systemic symptoms, impacting prognosis and survival [28,29], but also acting as markers to predict and monitor treatment responses [30]. Soluble interleukin-2 receptor (sIL-2R), interleukin-8 (IL-8) and immunoglobulin-free light chains (FLCs) are reported to be the most important predictors of outcome in patients with PMF with the activated JAK-STAT pathway [31].

The role of microenvironment in MPNs pathogenesis is also demonstrated by the high levels of tumor necrosis factor TNF-α and IL-6 detectable in mice injected with *JAK2V617F*-transduced BaF3 cells. Importantly, the anti-JAK2 miRNA leads only to a partial inhibition of IL-6 and IL-1, supporting that they are mostly released by stromal cells, rather than mutated cells [32,33]. The evidence of immunological dysregulation in PMF derives also from the murine Gata1^low^ mouse model, which shows a high level of TGF-β1 and collagen in megakaryocytes and abnormal signature in TGF-β1 signaling gene expression in spleen and marrow [34,35,36]. The dysregulation of immune system in MPNs stems, in part, from the defective differentiation of monocytes into dendritic cells [37] and from their inability to produce and release cytokines, such as IL-1β, TNF-α, IL-6 and IL-10, in response to infectious stimulus and to migrate into damaged tissues [38,39].

Furthermore, the link between chronic inflammation, fibrosis and clonal proliferation arises from a recently published work in which the authors show the key role of the chemokine CXCL4/platelet-factor-4 (PF4) in the fibrosis development through activation of profibrotic pathways and, notably, once again, of the JAK/STAT pathway [40].

One of the most convincing proofs that the inflammatory state is to be considered a driver of disease from the very beginning—and even in the most indolent forms—derives from some experiments conducted on patients with essential thrombocythemia (ET) and polycythemia vera (PV). In the works of Durmus et al., in fact, parameters such as serum total antioxidant status (TAS), total oxidative status (TOS), malondialdehyde (MDA) and oxidative stress index (OSI) were measured before and after the standard treatment for these diseases, detecting a corrective effect of the latter on the pathological alterations of the values [41,42]. Thrombotic events appear to be related in particular to the TOS. Furthermore, excessive ROS levels appear to confer a propensity for progression to PMF or AML [43].

Additionally, the link between chronic inflammation and clonal evolution has been substantiated by the findings that TNF-α facilitates the clonal expansion of *JAK2V617F*-positive cells in MPNs [44]. 

Whole-blood transcriptional profiling studies in primary myelofibrosis revealed a significant deregulation of interferon-inducible gene 27 (IFI27) [45], together with other genes that modulate inflammatory processes or involved in immunoregulation [46]. 

Once again, to reiterate the concept of the “biological continuum” mentioned other times in this discussion, transcriptional studies revealed significant deregulation of genes involved in immunoregulation and unequivocally link the three “classic” MPNs, from the most indolent (essential thrombocythemia) to conditions with a poorer prognosis such as overt myelofibrosis. Although there are elements that distinguish the genetic patterns of ET, PV and PMF, certain clusters of gene ontology terms support the interconnectedness of the different phenotypes, and the possibility that more indolent conditions transform into more aggressive forms. This does not imply that the natural history of these diseases is necessarily characterized by an onset in forms with a low prognostic impact, and that in any case myelofibrosis is expected as the last evolutionary step. Indeed, it is undeniable that ET, PV and PMF can have an autonomous and independent origin. Furthermore, if on the one hand we observe—clinically and biologically—transitional forms, it is equally true that the risk of fibrotic progression of a pure ET is relatively low. Myelofibrosis, whether primary, post-PV or post-ET, would still be the result of greater genetic instability and complexity, with the latter potentially responsible for a leukemic evolution [46,47]. 

Moreover, gene expression profiles in MPN clearly show upregulation of inflammation-related genes; particularly, variable expression in systemic inflammation and toll-like receptor signaling are significantly different between prefibrotic and overtly fibrotic MPN [48].

## 3. Inflammasomes and Their Determining Role in MPNs

Studies on the triggering of inflammatory processes in malignant hematopoiesis, with particular regard to MPNs, are focusing on the role of the innate immune system. The NLRP3 and AIM2 inflammasomes, which can recognize molecular patterns associated with pathogens and nuclear material of apoptotic cells, are mainly responsible for maintaining the cytokine storm, symptoms and end-organ damage that this group of diseases carry with them [49,50].

Inflammasomes are a class of cytosolic complexes of protein leading to the activation of potent inflammatory response, since they promote the expression, maturation and release of proinflammatory cytokines, such as IL-1β, to engage innate immune defenses [51]. Monocytes, macrophages, neutrophils and dendritic cells recognize pathogen-associated molecular patterns (PAMPs) or danger-associated molecular patterns (DAMPs) by nucleotide-binding oligomerization (NOD)-like receptors (NLRs), a type of pattern-recognition receptors (PRRs) including toll-like receptors (TLRs), C-type lectins (CTLs) and galectins [51,52]. The recognition of DAMPs/PAMPs by PRRs causes the assembly of high-molecular caspase platforms [53,54]. Following their activation, the canonical inflammasomes could process zymogen procaspase-1 into the active noncovalently linked subunits p10 and p20 (active caspase-1) leading to the maturation and release of proinflammatory cytokines (pro-IL-18 and pro-IL-1β) and inducing the formation of pores on plasma membranes [53].

IL-1β plays a key role in modulating the expression of genes responsible for fever, vasodilation and hypotension and allows innate immune cells to infect or damage tissues. On the other hand, IL-18 mediates adaptive immunity and it is fundamental for interferon-gamma (IFN-γ) production [55]. 

The nucleotide-binding domain, leucine-rich-containing family, pyrin domain containing 3 (NLRP3) inflammasome is the best characterized innate immune sensor protein complex. NLRP3 inflammasome—also called cryopyrin—is codified by the NLRP3 gene located on chromosome 1 and expressed in several cells involved in innate immune response, including monocytes, neutrophils, lymphocytes, epithelial and endothelial cells. It consists in a N-terminal pyrin (PYD) domain, a central nucleotide-binding oligomerization (NOD or NACHT) domain, and a leucine-rich repeat (LRR) domain at the C terminus [56]. After the recognition of pathogens and other damage-associated signals, NLRP3 protein interacts with the protein adaptor ASC by its pyrin domain and with a pro-caspase-1. The NLRP3 NOD domain has ATPase activity required for NLRP3 oligomerization following activation [57]. NLRP3 is involved in both sterile and non-sterile inflammation triggers however its specific mechanisms are still under study [58]. In recent times, it has been called into question as the most likely pathogenetic model of acute lung injury in coronavirus disease 2019 (COVID-19) [59].

As previously described by M.G. Netea et al., in macrophages and dendritic cells, two stimuli are needed for NLRP3 activation [60]. The first one, also called priming step or signal 1, is provided by inflammatory stimuli such as TLR4 agonists which induce NF-κB-mediated NLRP3 and pro-IL-1β expression; the second (activation step or signal 2) is triggered by PAMPs, DAMPs or glucose and amino acids efflux, thereby promoting NLRP3 inflammasome assembly and caspase-1-mediated IL-1β and IL-18 secretion and pyroptosis (Figure 2A) [61]. Pyroptosis is a recently recognized process of inflammatory programmed cell death, characterized by peculiar features, even if similar to apoptosis for some aspects it is quite peculiar for others, in particular for its form of DNA damage induction. Pyroptosis, in addition to having been associated with some vascular risk conditions such as atherosclerosis and diabetes, also seems to influence tumor growth and its aggressiveness. The mechanism is caspase- (and thus inflammasome-) dependent, and the final effector is the gasdermin D (GASD), a substrate of caspases, capable of forming pores in the plasma membrane. This results in gross membrane defects, and the release of proteins from the cytosol and cytokines [62]. 

In addition to the mechanisms involved in NLRP3 inflammasome activation—as above mentioned—several studies demonstrated how NLRP3 activity is also regulated by multiple post-translation modifications [63], such as ubiquitination [64,65] and phosphorylation [66,67] and by multiple NLRP3-interacting proteins including the molecular chaperone heat shock protein 90 (Hsp90) and its co-chaperone SGT1 [68], thioredoxin-interacting protein (TXNIP), guanylate-binding protein 5 (GBP5), double-stranded RNA-dependent protein kinase (PKR), migration inhibitory factor (MIF), microtubule-affinity regulating kinase 4 (MARK4) and Nek7 [69]. 

Furthermore a functional link between *KRASG12D* mutation and NLRP3 activation by the KRAS/RAC1/ROS/NLRP3 axis has been demonstrated both in *KRAS* mutant murine bone marrow (BM) and in leukemia human cells [70]. 

Important to be noticed is NF-KB, a central transcription factor involved in inflammation and immunity processes. NF-KB target genes include cytokines and chemokines leading to a positive feedback loop given that its functionality is also induced by inflammatory cytokines such as TNF-α or IL-1 besides LPS or other non-canonical activators [71]. 

The role of NF-KB signaling in pathogenesis of MPNs has been partially elucidated by M. Kleppe et al., demonstrating that NF-KB is constitutively active in mouse models independently from the presence of *MPLW515L* mutation and suggesting that its activity is modulated by Bromodomain and extraterminal domain (BET) proteins. Indeed the use of BET inhibitors leads to a decrease of proinflammatory cytokines production [72]. Moreover, Zhou et al. demonstrated that expression of NF-κB1, NLRP3 and IL-1β were higher in BM cells of patients harboring *JAK2V617F* mutation and splenomegaly underlying the cross talk between JAK2 and NF-KB. Notably the NF-κB-94 ins/del ATGG (rs28362491) polymorphism is linked to the increasing of the expression of NF-κB1 and NLRP3 identifying a potential therapeutic target in MPN neoplasms [73]. 

Another inflammasome that has aroused considerable interest in the context of MPNs is the so-called absence in melanoma 2 (AIM2). It too can recruit and activate caspase-1, but its main feature is that it is equipped with a sensor for apoptotic dsDNA (Figure 2B) [58]. After binding with a dsDNA of at least 80 base pairs by its positive charged hematopoietic expression, interferon-inducible nature, and nuclear localization (HIN) domain [74], the functioning of AIM2 is similar to NLRP3, being able to convert ASC into its prion form, favoring the interactions of its pyrin domains (PYD). The other domain (CARD) is instead offered for interactions with the analogous domain of caspase-1, which can; thus, be recruited and activated [75]. 

Importantly, an experiment conducted by Liew et al. allowed us to understand how multipotent hematological cell lines expressing *JAK2V617F* are characterized by the ability to trigger AIM2. The cell line produced, called D9, not only showed a proliferation that did not depend on hematopoietic growth factors (which are usually even essential for survival), but presented a higher gene expression of AIM2, CASP1, and IL-1β. This mechanism is clearly among those responsible for bone marrow fibrosis [76]. The activation of AIM2 relates myelofibrosis and systemic lupus erythematosus (SLE), since in both cases the aberrant macrophage maturation is related to the role of this inflammasome as a sensor of dsDNA deriving from apoptotic cells [77].

Finally, the combined action of NLRP3 and AIM2 justifies the acceleration of atherosclerosis processes in these diseases: while the first recognizes cholesterol crystals, the second is stimulated by cell death [78,79].

## 4. MPN and Autoimmune Diseases: Similarities and Correlations

The hyperactivation of the innate immune system through gross stimuli and the consequent assembly of scaffold proteins that lead to cytokine hyperproduction seems to be the pivot around which chronic myeloproliferative diseases and rheumatic diseases revolve—and often converge. The connection is so strong that today it is legitimate to speak of predispositions of one condition to another and coexistence of clonal and oligoclonal forms of inflammatory disease.

A deregulated inflammasome activity has in fact been reported in neuroinflammatory diseases [58] such as Parkinson’s disease and (PD) and Alzheimer’s disease (AD) [80] or multiple sclerosis (MS) [81], in metabolic diseases [82], inflammatory disorders [83] and in autoimmune rheumatic diseases including rheumatoid arthritis (RA), systemic lupus erythematosus (SLE), ankylosing spondylitis (AS) and Sjögren’s syndrome (SS) [84]. 

From an epidemiological point of view, a population-based study by the Swedish group of Kristinsson et al. is particularly important. It included 11,039 patients with MPNs and 43,550 matched controls and aimed to associate a personal history of autoimmune disease (selected broad spectrum) and the incidence of myeloproliferative neoplasms. Many pre-existing autoimmune conditions (including unsuspectedly immune thrombocytopenia) were associated with a higher risk of developing MPN, albeit with a sometimes limited statistical value [85].

Inspired by the aforementioned work, a group of researchers from the University of Pisa retrospectively analyzed over 5000 clinical records (2009–2019), identifying 55 patients with myelodysplastic syndrome (MDS, 20/55) or MPNs (35/55) associated with an autoimmune disease. Patients with MPNs are more frequently affected by PMF, and the hematological diagnosis usually precedes that of autoimmunity, contrary to what has been observed for MDS. Rheumatoid arthritis and antiphospholipid antibody syndrome are associated with PMF, while connective tissue disorders and anti-Ro52 positivity with ET [86]. 

Surely, the cytokine storm and the consequent clinical manifestations are intriguing aspects for both MPNs and AID, and the biological functions of the various proinflammatory molecules are manifold. But in this review, we would like to emphasize more than anything else the beginning and the end of the story, which in this case are the trigger of the inflammatory state and terminal damage. These two aspects allow us to understand the deep relationship between MPNs and AID. This can have non-trivial consequences in terms of therapies.

The first aspect—the initial one—is related to a genetic polymorphism. The JAK2 46/1 haplotype, in fact, is found to be the most frequently associated with the acquisition of the *JAK2V617F* point mutation and the onset of an MPN. There is therefore a model in which a constitutionally determined predisponent factor turns out to be a determinant of a specific somatic mutation. In the specific context, however, the demonstration that the aforementioned haplotype is frequently found in auto-inflammatory pathologies, and in particular in Crohn’s disease, is of considerable interest.

The second element is constituted by the fibrosis of the bone marrow. According to the studies by Ciaffoni et al., this is due to a particular pattern of gene expression, dominated by a non-canonical TGF-β signaling. The same signaling is responsible for the fibrotic processes in autoimmune diseases. Furthermore, the same group of researchers found that the plasma levels of anti-mitochondrial antibodies in PMF patients were certainly higher than that of controls, in accordance with the amount of circulating mitochondrial DNA [87,88].

Although the observation has been confirmed, there is an apparent paradox if we consider the relationship between IL-1β and TGF-β and their function as antagonists in the processes of hematopoiesis and immunoregulation. While the first cytokine stimulates the growth and differentiation of hematopoietic cells, modulating genes also belonging to elements of the stroma, TGF-β counteracts its actions by making the IL-1β receptors disappear on cell surfaces and stimulating the production of receptor antagonists [89]. Speculatively, we can hypothesize that—in the face of an overproduction of IL-1β found in both PMF and AID—in PMF there is an imbalance towards TGF-β while in AID this does not happen completely. This would also affect the T helper structure, more shifted towards a Th17 phenotype in AID.

Moreover, autoinflammatory disorders such as Mediterranean fever, familial cold autoinflammatory syndrome, Muckle-Wells syndrome, neonatal onset multisystem inflammatory disease (NOMID), hyperimmunoglobulin D syndrome and adult-onset Still disease generated much interest regarding the opportunity to treat patients affected by these syndromes using IL-1 receptor antagonist (IL-1Ra) or monoclonal anti–IL-1β antibodies. Since blood monocytes from autoimmune disorders patients release a higher quantity if IL-1β than controls, it is not excluded that these may be sensitive targets even in the early and hyperproliferative phases of MPNs [60]. 

Additionally, worthy of interest is the observation of a clinical-pathological entity of non-clonal origin called “autoimmune myelofibrosis”, which can be associated with flourishing and overt phases of autoimmune diseases. The histological aspects are almost entirely MPN-like, with mild fibrosis and clusters of megakaryocytes, but with a more frequent finding of lymphocyte aggregates. More or less marked cytopenias may be present, but splenomegaly is absent or very modest. In small series of cases, some response on fibrosis has been achieved with immunosuppressive therapy. During the follow-up, no aspects compatible with MPNs emerged, while infectious complications were recorded [90].

## 5. Inflammasomes and Iron Metabolism 

Although many pathogenetic features have been elucidated, the functional effects of iron in inflammasome activation in MPNs still needs to be clarified. In the last years, several scientific efforts have enormously expanded knowledge of iron metabolism focusing the attention on the imbalance between iron stored in the cells and the iron functionally available. The key protein implicated in iron regulation in the body is the hepcidin, an acute phase protein produced in the liver especially in inflammatory conditions. The immunological dysregulation and aberrant inflammatory cytokines production in MPNs cause an increase of hepcidin, a negative regulator of iron homeostasis, which reduces the bioavailable iron through inhibition of intestinal absorption, downregulation of the iron exporter channel ferroportin and increase of iron deposits in the monocyte-macrophage system. Anemia, that is the most common cytopenia in PMF, other than ineffective hematopoiesis, splenic entrapment, hemolysis or any bleeding losses, is partially secondary to this mechanism [91]. How the iron metabolism is linked to inflammasome activation is still under investigation. What is known is that cellular labile iron induces ROS production, which are able to release IL-1β in human monocytes through activation of NLRP3 and to induce hepcidin synthesis. This mechanism inevitably creates a positive feedback augmenting the condition of hyperinflammation. In a faithful reproduction, the mouse model with a myeloid lineage-specific ferroportin deficiency, produces high levels of TNF-α in response to LPS and show iron accumulation in reticuloendothelial macrophages of liver, spleen and bone marrow [92]. 

Since inflammasomes play a critical role for inflammation-associated organ fibrosis, it would be interesting to investigate the role of their (iron-induced) activation in PMF. Overall, activation of the NLRP3 inflammasome by cellular labile iron might have broad implications for iron overload and chronic inflammation in myeloproliferative disorders.

## 6. Inflammasome Activation and Hypercoagulable States

Inflammasomes have been ascribed as involved in the development of venous thromboembolism [93,94]. The tight correlation between inflammation and hemostasis activation is well known [95]. 

Any imbalance between both systems may lead to disease onset or development. Thrombosis derives from the abnormal, uncontrolled cross talk between inflammation and hemostatic systems. Inflammation seems crucial in thrombus development as it is involved in coagulation activation [95]. Coagulation is, on the other side, able to heighten inflammation [96]. Molecular mechanisms involved in this crosstalk are still under study. Inflammatory mediators, ATP gated ion channels, ROS released by mitochondria, Pannexin1 and P2X [97,98] and Pannexin1 and P2X purinergic receptor 7 (P2 × 7) are currently considered as NLRP3 activators [99]. In some immune cells like macrophages and dendritic cells a priming with cytokines, bacterial particles are needed for NLRP3 activation [100]. Interestingly, in platelets, NLRP3 is constitutively expressed, not requiring any priming or upstream activation [101]. Besides the classical Virchow’s triad, including hypercoagulability, hemodynamic changes and endothelial dysfunction, hypoxia also confers an increased risk of thromboembolic complications and represent the connection for activation of inflammasome complexes. Specifically, in this condition, hypoxia-inducible factor 1-alpha (HIF-1α) mediates NLRP3 activation and accelerates thromboembolic events [93]. Additionally, Yadav et al. have described in a mouse model that the activation of NLRP3 inflammasome and IL-1β induces thrombus formation even under normal oxygen concentration [94].

NLRP3 drives pyroptosis, a programmed cell death activated within an inflammatory context, by caspases. Pyroptosis consists in cell lysis accompanied by ruptured cell membranes. Inflammasome-mediated pyroptosis supports host defense against bacterial pathogens; however, an uncontrolled activation of pyroptosis exposes to the risk of disseminated intravascular coagulation (DIC), multiple organ failure (MOF) and death [102].

The tissue factor (TF) pathway is crucial in hemostasis and thrombo-inflammatory diseases [103]. TF released from macrophages during pyroptosis is able to activate coagulation and is thus associated with increased thrombotic risk. The most plentiful source of circulating TF secondary to inflammasome activation is constituted by monocytes and macrophages. TF released by monocytes and macrophages has been involved in DIC secondary to sepsis [104]. Inflammasome-generated caspase-1 mediates actin translocation across membranes thus enabling the last phase of thrombo-inflammatory extracellular vesicles (EV) shedding [105]. Endothelial cells can also undergo pyroptosis following specific stimulation with LPS [106].

Activated NLRP3 inflammasome upregulates caspase-1 activity and triggers the production of bioactive IL-1β and other proinflammatory cytokines, such as IL-18. NLRP3 inflammasome has been found in platelets activated by dengue virus [107]. Activation of platelets by collagen or thrombin is also able to upregulate NLRP3 inflammasome activation [101]. Platelets constitutively express NLRP3, ASC and BTK. BTK, a cytoplasmic tyrosine kinase expressed by B-cells, myeloid cells and platelets, has been recently shown to modulate NLRP3 inflammasome in immune cells [108] and platelets [101]. In detail, BTK gene deletion in platelets inhibited platelet activation, aggregation and thrombus formation, restored by NLRP3 activators.

Bakele et al. first showed that human neutrophils express several parts of the inflammasomes and release IL-1β cytokine upon activation [109]. The link between inflammasome activation in neutrophils, IL-1β release and the risk of venous thrombosis has been recently described in a mouse model in hypoxic conditions [93]. 

## 7. Selected Targeted Therapies

The biological alterations that involve the innate immune system and that unite MPNs to AID are fertile ground for designing new therapeutic strategies that directly or indirectly (through its pathways) affect the inflammasome. Below is a quick list of therapies selected by our team based on our experience and a possible short-term development.

(a)BET inhibitors (i.e., CPI-0610): A recent study identifies nuclear factor kB (NF-kB) signaling as a key pathway activated in malignant and non-malignant cells in MPNs. The authors demonstrated not only that its inhibition attenuates and reduces cytokine production in vivo, but, most importantly, that dual JAK/BET inhibition induces a marked reduction of inflammatory cytokines level and of the allele burden, and reverses bone marrow fibrosis in vivo [72]. CPI-0610 is currently being tested in a phase three trial in combination with ruxolitinib (NCT04603495).(b)IRAK4 inhibitors (i.e., CA-4948): Interleukin-1 receptor associated kinase (IRAK4) is a serine/threonine kinase involved in the regulation of innate immunity that is recruited by MyD88 following the activation of the toll-like receptor (TLR) and interleukin receptor, initiating a signaling cascade that induces cytokine and survival factor expression mediated by the transcription factor NF-KB. MyD88 and IRAK4 are key regulators of fibrosis development and inflammation; their interaction is a prerequisite for pathological fibrogenic response. In fact, the specific ablation of MyD88 in pericytes or pharmacological inhibition of MyD88 signaling by an IRAK4 inhibitor in vivo attenuates tissue injury, activation and differentiation of myofibroblasts [110]. CA-4948 is a small molecule inhibitor of IRAK4 kinase that modulates the TLR and IL-1R signaling cascades. CA-4948 is being developed as a novel agent for the treatment of hematologic cancers with dysregulated IRAK4 signaling.(c)BTK inhibitors: (i.e., TL-895) The Bruton’s kinase (BTK) is fundamental for NLRP3 activation, as one of its roles is to promote oligomerization and speck formation of ASC [111]. Therefore, inhibition of BTK strongly affects the NLRP3 inflammasome efficiency. Compounds like ibrutinib are also capable of reducing IL-1β production via the suppression of caspase-1. TL-895 is currently evaluated in clinical trials for the treatment of myelofibrosis.(d)“Not-only-JAK” inhibitors: Among the new JAK inhibitors, some of them target kinases other than JAK2. There is a potential additive anti-inflammatory effect. A common example is the inhibition of IRAK-1 and IL-1 related signaling by pacritinib [112].

## 8. Conclusions

The biological complexity of MPNs is still far from being revealed, but new insights into the role of inflammasomes are allowing the development of new treatment strategies. We believe that the near future is mainly represented by combination therapies that attempt to demolish the three pillars on which the pathogenesis and course of these diseases rest: uncontrolled proliferation, systemic inflammation and loss of immunoregulation. A multidisciplinary approach involving rheumatology experts, given the multiple points in common between MPNs and AID, would be absolutely desirable. Aware of the potential scarce applicability of this approach, as it is not yet universally accepted, it would be important for hematologists or oncologists to acquire a multidisciplinary competence for a broader and more correct interpretation of these diseases and their management. At the same time, translational research should examine the relationship that exists between clinical manifestations, risk scores, the molecular profile (and its evolution) and the involvement of inflammasomes, in order to build a personalized treatment proposal over time.

## Figures and Tables

**Figure 1 ijms-22-00561-f001:**
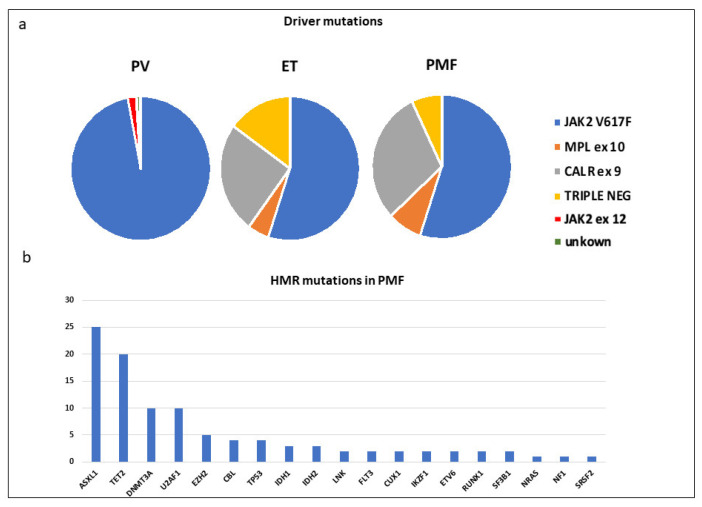
(**a**) Frequency of driver mutations in each MPN; (**b**) Frequency of high risk mutations in PMF as reported by Vainchenker and Kralovics [1]. MPN: Myeloproliferative Neoplasm; PV: Polycythemia vera; ET: Essential Thrombocythemia; PMF: Primary Myelofibrosis; HMR: high molecular risk.

**Figure 2 ijms-22-00561-f002:**
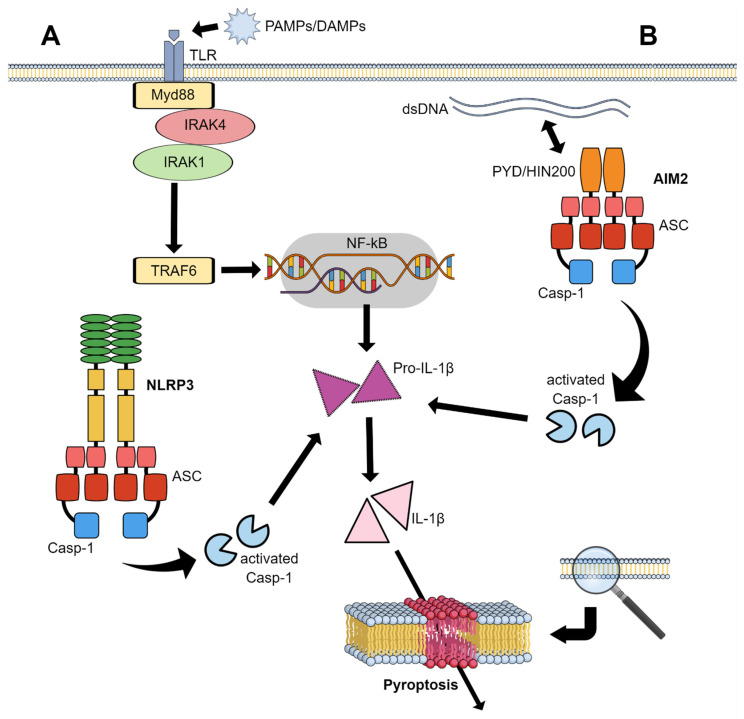
Pyroptosis induced by the release of proinflammatory cytokines, following their activation by caspase-1 (Casp-1). The latter is triggered by NLRP3 and AIM2 inflammasomes through two mechanisms: (**A**) For NLRP3, the interaction of pathogen-associated and damage-associated molecular patterns (PAMPs, DAMPs) with the toll-like receptor (TLR); (**B**) for AIM2, the recognition of apoptotic double strand DNA by the HIN domain.

## Data Availability

No new data were created or analyzed in this study. Data sharing is not applicable to this article.

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
