# Peer review of "Genetics and Pathogenetic Role of Inflammasomes in Philadelphia Negative Chronic Myeloproliferative Neoplasms: A Narrative Review"

_ijms, 2021, doi:10.3390/ijms22020561_

Round 1
Reviewer 1 Report
This is a comprehensive, up-to-date and elegantly written review on inflammation and immunity in myeloproliferative neoplasms.
Only few parts of the text deserve clarification:
In the paragraph “MPNs as inflammatory diseases”, the authors concluded that ”The progressive deregulation of certain clusters of gene ontology terms clearly delineated the evolution from ET and PV to PMF”. The authors cite papers form Skov et al. on functional analysis that suggest disease evolution from ET over PV to PMF. This reflects a trend of thinking that, simplifying the reality, states that MPNs represent an unique disease category with a visible or hidden natural evolution from ET to PV to MF. According to this idea, MF is the final step of a disease evolution. The results of gene expression profiling to which the authors make reference do not allow this conclusion. The continuum is a biological one but not a clinical one, and it does not deny an independent and autonomous origin of the three disorders. The risk of such a misunderstanding should be more clearly highlighted.
In the paragraph “Inflammasomes and their determining role in MPNs” the authors suggest a convergence of chronic myeloproliferative diseases and rheumatic diseases. They write “The connection is so strong that today it is legitimate to speak of predisposition of one condition to another and coexistence of clonal and oligoclonal forms of inflammatory disease”. After this sentence, the author should mention on the “autoimmune myelofibrosis” and discuss how this entity that is often associated with rheumatic/autoimmune disorders reinforce or deny the authors conclusion.
The authors conclude that ”A multidisciplinary approach involving rheumatology experts, given the multiple points in common between MPNs and autoimmune disorders, is absolutely desirable”. The authors should mention that this is not an universally accepted thinking and that myeloproliferative disorders represent an example of disorders for which the haematologist or oncologist needs to have a multidisciplinary competence.
Author Response
The reviewer's suggestions are elegant, insightful, and valuable. We are grateful for this posting.
Following an internal discussion, we decided to strictly follow the instructions to improve our manuscript.
We have therefore tried to eliminate the discrepancies and misunderstandings on the contrast between the biological continuum and the clinical onset of diseases, by reconsidering the article by Skov et al. and also by taking a cue from the words of the reviewer himself/herself.
We have added a small paragraph on autoimmune myelofibrosis. It was our intention to do so in the course of drafting, but we were hesitant. We are therefore pleased that it has been considered useful information.
A concept of multidisciplinarity that could be more "workable", or at least and more adapted to the absence of a universally shared point of view was expressed in the conclusions, as wisely suggested.
Reviewer 2 Report
Valeria Di Battista et al. systematically introduced the correlation of inflammation and MPN, the manuscript was organized well, and the context was convincing. I only have two small issues, here are my points:
- Please use the chart figure to show the ratio of genetic mutations in Part1.
- The role of proptosis in MPN is of interest to reviewer, I wish the authors can discuss more.
Author Response
We thank the reviewer for the opinion expressed and for the excellent suggestions. As indicated, we have provided further details on the mechanism of pyroptosis, and we have added charts on the frequencies of genetic mutations. We hope the changes are appreciable.
[Lines: 56; 74-75; 207-215]
Reviewer 3 Report
This manuscript summarizes the genetic and pathogenic role played by the inflammasome in myeloproliferative neoplasms (MPNs). To my view, it is an excellent narrative review with a high-impact in the field of MPNs. The paper is well-written and worthy of publication in the journal. The manuscript only requires minor revisions. Some suggestions for revision are listed below:
GENERAL RECOMMENDATIONS
- Explain all abbreviations at the first use in the text.
- Revise the references to match the style of the journal, e.g. the DOI is missing in many references, some references don’t list the issue, volume and page numbers etc.
SPECIFIC RECOMMENDATIONS
- Title: Since you are only discussing BCR-ABL1-negative MPNs, maybe it would be appropriate to add BCR-ABL1-negative MPNs in the title as well.
- In addition, I would suggest adding “a narrative review” in the title of the manuscript.
- “unequivocally link the three "classic" MPNs, with essential thrombocythemia.” – this sentence is a little bit ambiguous, aren’t PV, ET and PMF the three classic MPNs?
- Considering you have mentioned oxidative stress as a possible pathogenic mechanism in the development of MPNs, it would be interesting to add a short paragraph discussing the crosstalk between oxidative stress and inflammation in the pathogenesis of MPNs. I have selected some references which you can use if you find them suitable:
https://pubmed.ncbi.nlm.nih.gov/24642873/
https://pubmed.ncbi.nlm.nih.gov/24254553/
https://doi.org/10.37358/RC.19.10.7581
Author Response
We thank the reviewer for supporting our article and for his contribution to improving the final drafting.
We welcome the suggested title changes, which enhance the content of the manuscript. The sentence marked as "ambiguous" was definitely ambiguous! The concept we wanted to express was the transition from the most indolent forms to those with a poor prognosis. We have appropriately corrected.
We have also integrated a part on oxidative stress: the topic is of undoubted relevance, and addressing it in the part on inflammatory mechanisms is preparatory to the paragraph on iron metabolism. We found the suggested references very interesting.
[Lines: 1-2; 146-153; 161-162]